# Determining the Target Population That Would Most Benefit from Screening for Hepatic Fibrosis in a Primary Care Setting

**DOI:** 10.3390/diagnostics11091605

**Published:** 2021-09-03

**Authors:** Su Hyun Park, Jong Hyun Lee, Dae Won Jun, Kyung A Kang, Ji Na Kim, Hee Jin Park, Han Pyo Hong

**Affiliations:** 1Department of Internal Medicine, Hanyang University College of Medicine, Seoul 04763, Korea; suhyun.p@gmail.com; 2Department of Medical and Digital Engineering, Hanyang University College of Engineering, Seoul 04763, Korea; jonghyunlee1993@gmail.com; 3Department of Radiology, Samsung Medical Center Sungkyunkwan University School of Medicine, 81 Irwon-ro, Gangnam-gu, Seoul 06351, Korea; 4Department of Radiology, Kangbuk Samsung Hospital, Sungkyunkwan University School of Medicine, Seoul 03181, Korea; hebecrom@hanmail.net (J.N.K.); parkhiji@gmail.com (H.J.P.); 5Biostatistical Consulting and Research Lab, Medical research Collaborating Center, Hanyang University, Seoul 04763, Korea; dr.red9240@gmail.com

**Keywords:** non-alcoholic fatty liver disease, diabetes mellitus, fibrosis, population groups, referral and consultation

## Abstract

Due to its high prevalence, screening for hepatic fibrosis in the low-risk population is called for action in the primary care clinic. However, current guidelines provide conflicting recommendations on populations to be screened. We aimed to identify the target populations that would most benefit from screening for hepatic fibrosis in clinical practice. This study examined 1288 subjects who underwent magnetic resonance elastography. The diagnostic performance of the Fibrosis-4 (FIB-4) index and NAFLD fibrosis score was compared in the following groups: (1) ultrasonography (USG)-diagnosed NAFLD, (2) elevated liver enzyme, (3) metabolic syndrome, (4) impaired fasting glucose, and (5) type 2 diabetes regardless of fatty liver. Decision curve analysis was performed to express the net benefit of groups over a range of probability thresholds (Pts). The diabetes group showed a better area under the receiver operating characteristic curve (AUROC: 0.69) compared with subjects in the USG-diagnosed NAFLD (AUROC: 0.57) and elevated liver enzyme (AUROC: 0.55) groups based on the FIB-4 index. In decision curve analysis, the diabetes group showed the highest net benefit for the detection of significant fibrosis across a wide range of Pts. Patients with diabetes, even in the absence of fatty liver, would be preferable for hepatic fibrosis screening in low-risk populations.

## 1. Introduction

Chronic liver disease is common worldwide and has become a major global issue. The overall mortality rate increases as liver fibrosis progresses, up to 1.36-fold with greater than significant hepatic fibrosis [1]. Therefore, diagnosis should be achieved in early stages when liver fibrosis is mild-to-moderate. In addition, significant hepatic fibrosis has a prevalence of 5–7% in unselected populations [2,3]. Due to its high prevalence, screening for significant hepatic fibrosis in the low-risk population is called for action in the primary care clinic. However, screening for hepatic fibrosis in the general population is difficult in practice. 

A major obstacle for liver fibrosis screening in the general population includes the identification of groups to be screened. Although screening is not universally recommended, targeted screening in groups deemed at high risk for fibrosis remains a contentious issue, with conflicting guidance from major international societies. In most primary care settings, screening tests for liver fibrosis are focused on patients with suspicion of non-alcoholic fatty liver disease (NAFLD). However, only approximately 15–20% of patients with NAFLD develop non-alcoholic steatohepatitis (NASH) [4], the progressive form of liver disease. Therefore, the high prevalence and relatively low severity of NAFLD require the triage of patients for liver fibrosis screening. Meanwhile, despite the high prevalence of advanced fibrosis in patients with diabetes [3,5,6], consensus on the need for the screening of significant hepatic fibrosis in all diabetic patients is unclear, with a lack of evidence demonstrating benefit from screening. To the best of our knowledge, the cost-effectiveness of a screening strategy for NASH or advanced fibrosis in patients with diabetes has been evaluated in only two studies, which showed conflicting results [7,8]. Furthermore, in the screening strategies used in those studies, all patients first underwent ultrasonography (USG) to detect fatty liver. Whether fibrosis screening tests should be performed in all diabetic patients regardless of fatty liver remains unclear. 

Therefore, in the present study, the diagnostic performance of non-invasive tests for screening liver fibrosis based on various conditions including diabetes with and without fatty liver was evaluated to identify target groups that might most benefit in clinical practice.

## 2. Materials and Methods

### 2.1. Study Design

This was a cross-sectional study and included a subset of the Kangbuk Samsung Health Study (KSHS, Seoul, Korea) participants 18 years of age or older who underwent health screening examinations including magnetic resonance elastography (MRE) between January 2015 and May 2018 (*n* = 2149). The KSHS is a Korean cohort study including individuals who have participated in a comprehensive health screening examination annually or biannually at Kangbuk Samsung Hospital Total Healthcare Centers in Seoul and Suwon, South Korea. More than 80% of participants were employees of various companies or their family members. In South Korea, the Industrial Safety and Health Act requires health examinations for all employees. Examinations are offered free of charge, and MRE is voluntarily chosen by employees and paid for by their employer. This study was approved by the Kangbuk Samsung Hospital Institutional Review Board (IRB No. KBSMC 2020-11-029, approved on 26 November 2020).

### 2.2. Inclusion and Exclusion Criteria

To provide data on applicability of non-invasive tests in low-risk populations, participants were excluded if they had positive viral hepatitis serology (hepatitis B surface antigen (*n* = 167) or hepatitis C antibody (*n* = 12)), chronic liver disease caused by significant alcohol consumption (>20 g/day, *n* = 642) [9,10], or liver cirrhosis on USG (*n* = 27). Subjects who had incomplete data were also excluded (*n* = 13) (Figure 1).

### 2.3. MRE Measurement 

MRE was used as the reference standard for fibrosis. Although liver biopsy is the gold standard method to assess fibrosis, MRE is non-invasive, has no significant side effects, and is used in clinical practice as the reference standard to avoid liver biopsy. MREs were performed using 1.5T MRI systems (Signa HDxT; GE Healthcare in Seoul, and Optima 360 Advance, GE Healthcare in Suwon, Korea). Measurements were performed according to the manufacturer’s instructions. A standard 60-Hz shear-wave was generated by an acoustic passive driver attached to the body wall anterior to the liver and coupled with an acoustic active driver outside the scanning room. A two-dimensional gradient-recalled echo pulse sequence synchronized to the shear-wave frequency was acquired to obtain three noncontiguous axial slices (8-mm-thick sections). The wave images from each slice location were processed automatically using an inversion algorithm to generate axial liver stiffness maps. The image analyst drew regions of interest in parts of the liver showing wave propagation on the images, avoiding large blood vessels and artifacts. The mean stiffness value was calculated by averaging all pixel values, and the results were displayed automatically on each map in units of kilopascal (kPa). The fibrosis stages were defined as ≥ F2 (significant fibrosis) and ≥ F3 (advanced fibrosis), with thresholds of 3.0 and 3.6 kPa, respectively [11].

### 2.4. Clinical Assessment

Subjects were required to complete a structured questionnaire involving a survey of demographics, medical history, comorbid conditions, current alcohol consumption, and drug use. Subjects underwent complete clinical examination with laboratory tests. BMI was calculated as weight (kg) divided by height (m^2^). Waist circumference was measured at the midpoint between the lowest rib and top of the iliac crest. Blood pressure (BP) was measured by trained nurses using an automatic tonometer after 5 min of rest. Hypertension was defined as systolic BP ≥ 140 mmHg, diastolic BP ≥ 90 mmHg, or current use of antihypertensive medication. Diabetes was defined as fasting plasma glucose level ≥ 126 mg/dL, glycated hemoglobin (HbA1c) level ≥ 6.5%, and/or current use of antihyperglycemic medications. Impaired fasting glucose was defined as a fasting serum glucose level of 100–125 mg/dL [12]. Metabolic syndrome was diagnosed if three or more of the following risks were present [13]: (a) waist circumference ≥ 90 cm in males or ≥ 80 cm in females (in accordance with the International Obesity Task Force criteria for the Asia-Pacific population); (b) triglyceride level ≥ 150 mg/dL; (c) high-density lipoprotein (HDL) cholesterol level < 40 mg/dL in males and < 50 mg/dL in females; (d) blood pressure level ≥ 130/85 mm Hg or current use of antihypertensive medications; (e) fasting blood glucose level ≥ 100 mg/dL or current use of antidiabetic medications. Participants were considered to have NAFLD when they fulfilled all inclusion criteria and showed fatty liver on USG.

### 2.5. Abdominal USG

Abdominal USG was performed by experienced radiologists. Semi-quantitative grading of fatty liver was performed based on standard criteria [14]. Fatty liver status was defined as presence versus absence. 

### 2.6. Target Populations 

The subjects were divided into the following five groups: (1) USG-diagnosed NAFLD; (2) elevated liver enzyme (aminotransferases), defined as ≥1.5-fold the upper limit of normal; (3) metabolic syndrome; (4) impaired fasting glucose; and (5) type 2 diabetes regardless of fatty liver. 

### 2.7. Indices of Hepatic Fibrosis

The European Association for the Study of the Liver (EASL) and Italian guidelines suggest the use of NFS and FIB-4 index as non-invasive methods to identify patients with different risks of advanced fibrosis [4,15]. Furthermore, American Association for the Study of Liver Disease (AASLD) guidelines report a recent study in which both NAFLD fibrosis score (NFS) and Fibrosis-4 (FIB-4) index showed the best predictive value for advanced fibrosis among histologically proven NAFLD patients compared with other tests [16]. Therefore, single use of these two non-invasive tests was evaluated. FIB-4 index score ≥ 0.92 and NFS ≥ −3.16 based on the following formulas were used as the cut-off values for excluding significant hepatic fibrosis [17,18]:

FIB-4 [19]:(1)$$\frac{\mathrm{age}\;\left[\mathrm{years}\right]\times \;\mathrm{AST}\;\left[U/L\right]}{\mathrm{platelet}\;\mathrm{count}\;\left[109/L\right]\times \;\mathrm{ALT}1/2\;\left[U/L\right]}$$

NFS [20]:−1.675 + 0.037 × age [years] + 0.094 × BMI [kg/m^2^] + 1.13 × impaired fasting glycemia or diabetes [yes = 1, no = 0] + 0.99 × AST/ALT ratio − 0.013 × platelet [×10^9^/L] − 0.66 × albumin [g/dL](2)

### 2.8. Decision Curve Analysis and Net Benefit 

Test performance has been reported in the literature as area under the receiver operating characteristic curve (AUROC), sensitivity, and specificity. However, these measures do not determine whether a test is fit for clinical use.

In the present study, decision curve analysis was used to compare and describe the clinical effects between subgroups. Decision curve analysis involves calculating the net benefit, which places benefits and harms on the same scale and is presented in unit of true positives [21]. The cost component is not included in the analysis. Net benefit can be calculated using the following formula:(3)$$\mathrm{Net}\;\mathrm{Benefit}=\frac{\mathrm{TP}}{N}-\frac{\mathrm{FP}}{N}\left(\frac{\mathrm{Pt}}{1-\mathrm{Pt}}\right)$$
(TP: true positive, FP: false positive, N: total sample size, Pt: probability threshold)

The Pt is defined by considering the number of patients a clinician would need to perform a test on to obtain one correct diagnosis. For example, if a clinician found it acceptable to subject 10 patients to liver biopsy to obtain one correct diagnosis of cirrhosis, then Pt is 10%. A strategy is considered to have clinical value if it has the highest net benefit across a range of Pts.

### 2.9. Statistical Analysis

For descriptive statistics, results were expressed as mean ± standard deviation (SD) or median with interquartile range (IQR) for continuous variables with normal or asymmetric distribution, respectively. Categorical variables were presented as numbers and percentages. To compare variables based on gender, Chi-square test was used for categorical variables, and Student’s *t*-test was used for continuous data. To assess the performance of the FIB-4 index based on the screening algorithm to detect significant fibrosis, AUROCs were determined. IBM SPSS Statistics version 23.0 (IBM Co., Armonk, NY, USA) was used, and *p* < 0.05 was considered statistically significant.

## 3. Results

### 3.1. Baseline Characteristics

The clinical data of subjects are summarized in Table 1. The mean age of the subjects was approximately 50 years, and there were more males (77.2%) than females. The mean body mass index (BMI) was 24.58 ± 3.08. Among the 1288 subjects, 169 (13.12%) had diabetes, 418 (32.45%) had hypertension, and 317 (24.61%) had metabolic syndrome. The proportion of significant hepatic fibrosis with 3.0 kPa or higher based on MRE criteria was 3.88%, and the rate of advanced hepatic fibrosis with 3.6 kPa or higher was 0.47%.

### 3.2. Prevalence of Significant Fibrosis Based on Various Conditions

The prevalence of significant hepatic fibrosis was evaluated based on five conditions (USG-diagnosed NAFLD, elevated liver enzyme, metabolic syndrome, impaired fasting glucose, and diabetes); the prevalence of significant hepatic fibrosis (≥3.0 kPa) was 5.9%, 6.7%, 7.6%, 3.5%, and 14.2%, respectively. The prevalence of significant hepatic fibrosis in diabetic patients with USG-diagnosed fatty liver was significantly higher than in diabetic patients without fatty liver (6.67% vs. 20.2%, *p* = 0.027). A difference was not observed in the prevalence of significant hepatic fibrosis in patients with elevated liver enzyme, metabolic syndrome, or impaired fasting glucose based on the presence of fatty liver on USG (elevated liver enzyme; 3.91% vs. 8.66%, *p* = 0.115, metabolic syndrome; 2.68% vs. 4.84%, *p* = 0.158, and impaired fasting glucose; 2.46% vs. 4.69%, *p* = 0.20).

### 3.3. Performance of FIB-4 Index and NFS in Screening for Significant Hepatic Fibrosis in Various Conditions

Table 2 summarizes sensitivity, specificity, positive predictive value (PPV), and negative predictive value (NPV) for the FIB-4 index and NFS based on various conditions. All groups had high NPV (>90%) for the exclusion of significant fibrosis. Notably, the diabetes group showed the best PPV for both the FIB-4 index and NFS among the groups (17.7% and 14.7%, respectively). For the FIB-4 index and NFS, the PPV was 6.9% and 6.2% in the USG-diagnosed group and 6.2% and 6.5% in the elevated liver enzyme group, respectively. The sensitivity for identifying significant fibrosis in patients with diabetes was low and similar for the FIB-4 index and NFS (40.0% and 46.0%, respectively). 

The use of the FIB-4 index resulted in a 6.9% true positive rate and 93.1% false positive rate in the USG-diagnosed NAFLD group (Figure 2). In the diabetes group, the FIB-4 index showed a true positive rate of 17.7% and false positive rate of 82.3% (Figure 2). 

In the diabetes group, the FIB-4 index performed better than the NFS (AUROC: 0.69 vs. 0.58, *p* = 0.05) (Figure 3). However, the performance of these two methods was not significantly different in the other subgroups.

### 3.4. Decision Curve Analysis and Net Benefit

The diabetes group showed the highest net benefit compared with the USG-diagnosed NAFLD and elevated liver enzyme groups for the detection of significant fibrosis across a wide range of Pts (Figure 4, Appendix A). The net benefit for each group was lower than for the strategy of “refer all” for Pt below approximately 0.03. At Pt 0.07, where an FIB-4 index-based strategy provides greater net benefit than liver biopsy [22], the diabetes group showed the greatest net benefit, followed by metabolic syndrome. Notably, net benefit was not apparent when using a Pt of 0.07 for patients with USG-diagnosed NAFLD. At the traditional Pt of 0.2 (five or fewer referrals for each correct diagnosis of fibrosis), a positive net benefit was not observed in all groups.

## 4. Discussion

In the majority of previous studies, hepatic fibrosis screening was assessed to determine the optimal diagnostic tests in the referral process. To the best of our knowledge, no prior studies investigated who should undergo screening for hepatic fibrosis in the community setting. In the present study, which target population in a low-risk cohort might most benefit from screening for hepatic fibrosis was determined. The patients with diabetes showed better performance (AUROC: 0.69) with the best combination of PPV and NPV (17.7% and 97.5%, respectively) compared with the USG-diagnosed NAFLD (AUROC: 0.57) and elevated liver enzyme (AUROC: 0.55) patients using the FIB-4 index-based algorithm. The diabetes group showed a 2.5-fold increase in the detection of significant fibrosis (7% to 18%) and simultaneously reduced the percentage of unnecessary referrals (93.1% to 82.3%) compared with the USG-diagnosed NAFLD group (Figure 5). In decision curve analysis, the diabetes group showed the highest net benefit compared with the USG-diagnosed NAFLD and elevated liver enzyme groups for the detection of significant fibrosis across a wide range of Pts. Taken together, these results indicate a possible rationale for the targeted screening of patients with diabetes.

It is very well known that screening for hepatic fibrosis is necessary in diabetes or NAFLD cohorts. In this study, we tried to compare the size of net benefits for hepatic fibrosis screening among known various high-risk groups in community cohorts. So far, a few social economic analyses were performed in selected high-risk populations (NAFLD cohort or diabetes cohort) with conflicting results [7,8], but a comprehensive net benefit analysis according to various high-risk groups in primary care settings has not been reported. We believe that the results of this study can be helpful in understanding the target population and priority of screening algorithms in various high-risk groups in primary care centers.

Because awareness regarding the health risks associated with hepatic fibrosis in patients with diabetes is increasing, there is renewed interest in screening for hepatic fibrosis in diabetic patients. In this context, the feasibility of non-invasive tests for hepatic fibrosis screening in patients with diabetes was evaluated in two recent studies [23,24]. The results showed intermediate performance (AUROC of FIB-4 index: 0.70 and 0.78), which is similar to slightly higher compared with our results (AUROC: 0.69). However, these validation cohorts included diabetic patients with NAFLD and did not properly validate a diabetic cohort regardless of fatty liver. As the effects of diabetes and fatty liver might be additive in the same individual, diagnostic performance derived from cohorts with high fibrosis prevalence might have been overestimated in the previous study. The presence of diabetes is associated independently with liver fibrosis [3,25]. Therefore, to accurately estimate the performance of non-invasive tests in patients with diabetes, subjects without fatty liver need to be included in the validation group.

In the present study, the FIB-4 index had better performance for screening significant hepatic fibrosis in the diabetes group than did the NFS (Figure 3). Consistent results were found in two previous studies [23,24]. This finding is clinically relevant because the NFS, different from the FIB-4 index, includes the presence of diabetes in the formula. This indicates that the diagnostic performance of the NFS could be significantly influenced in a cohort of diabetic patients. Although several international guidelines recommended the FIB-4 index or NFS as the first-line examination for detecting fibrosis in NAFLD patients, the FIB-4 index is preferable in diabetic clinics.

In the present study, decision curve analysis was used to quantify the harms and benefits of using two simple non-invasive fibrosis tests based on various conditions rather than only presenting the diagnostic test accuracy. Fortunately, the diabetes group provided the greatest diagnostic utility in a low-risk population. 

The present study had several limitations. First, for diagnosing liver fibrosis, biopsy remains the gold standard. However, performing a biopsy in asymptomatic participants in this low-risk cohort would have been unethical. Therefore, MRE was used, which is currently considered the most accurate imaging tool for the detection of liver fibrosis. Second, only the FIB-4 index and NFS were used, without considering other available non-invasive tests (e.g., AST/ALT ratio, AST to platelet ratio index, vibration-controlled transient elastography) or a combination of tests. However, the FIB-4 index and NFS are the most widely used and best validated tests. In addition, the main objective was to compare the diagnostic performance between groups rather than providing AUROC. Third, modified thresholds of the FIB-4 index and NFS used to predict significant fibrosis are not properly validated yet. The FIB-4 index and NFS were originally proposed to exclude advanced fibrosis using the cut-off values of 1.3 and −1.455. It would be inappropriate to use 1.3 and −1.455 as a lower cut-off in the primary care setting as it would include significant fibrosis. These modified cut-off values need to be validated in a large cohort in a low-risk population. Additionally, a reasonable range of Pts is determined by the preference of clinicians or patients and is not objective. However, rank order between groups irrespective of Pt was provided and demonstrated the highest net benefit in the diabetes group. The cohort in this study comprised relatively young and healthy individuals, and whether these results can be generalized to the elderly population is unclear. Finally, financial costs were not evaluated, which is necessary to apply research results in clinical practice. However, model-based economic evaluation is afflicted with certain limitations because an unrealistic simplification of assumptions is required.

## 5. Conclusions

In conclusion, patients with diabetes, even in the absence of fatty liver, would be preferable for hepatic fibrosis screening in a primary care setting. However, the cost effectiveness remains unclear and should be addressed in future studies. 

## Figures and Tables

**Figure 1 diagnostics-11-01605-f001:**
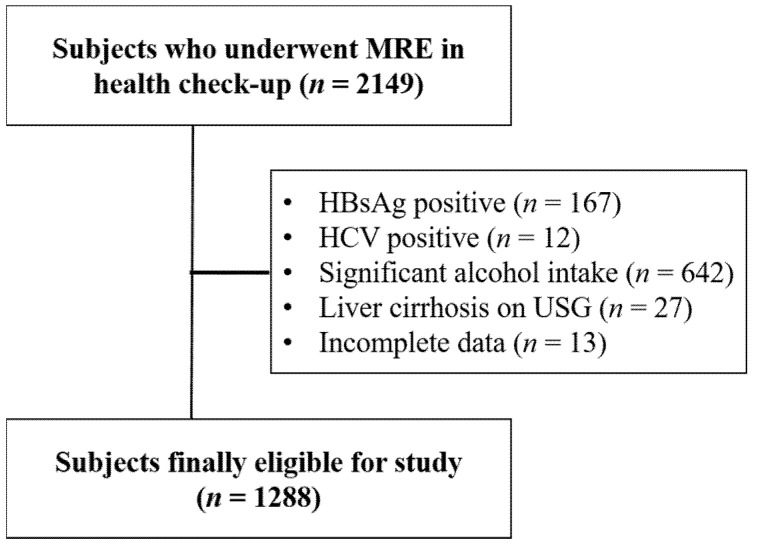
Flow chart of study participants. MRE, magnetic resonance elastography.

**Figure 2 diagnostics-11-01605-f002:**
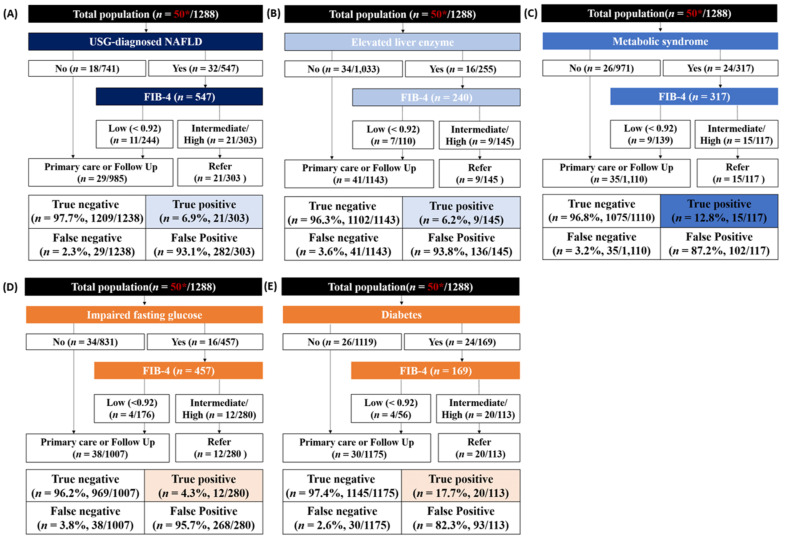
Test outcomes and referral rates of FIB-4 index-based screening in five conditions. (**A**) USG-diagnosed NAFLD; (**B**) Elevated liver enzyme; (**C**) Metabolic syndrome; (**D**) Impaired fasting glucose; (**E**) Diabetes. * The number of patients with significant fibrosis with 3.0 kPa or higher based on MRE criteria.

**Figure 3 diagnostics-11-01605-f003:**
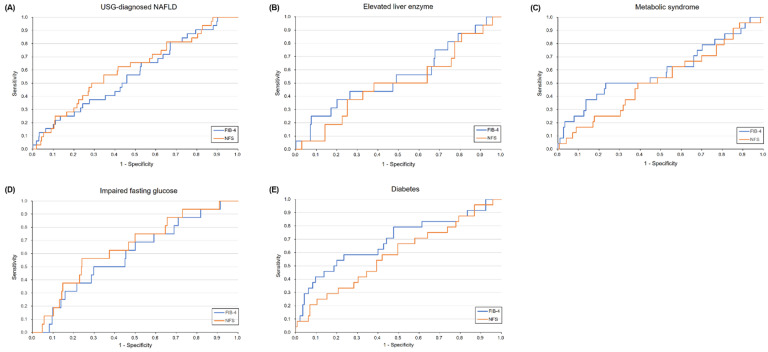
Diagnostic performance of FIB-4 index and NFS for diagnosis of significant fibrosis in five conditions. (**A**) USG-diagnosed NAFLD; (**B**) Elevated liver enzyme; (**C**) Metabolic syndrome; (**D**) Impaired fasting glucose; (**E**) Diabetes.

**Figure 4 diagnostics-11-01605-f004:**
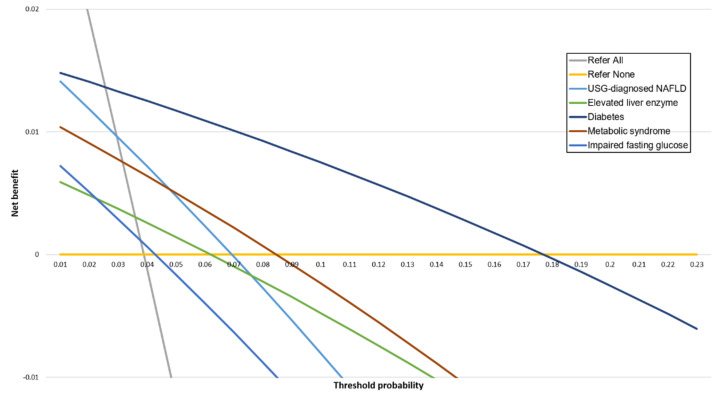
Decision curves for FIB-4 index in five groups.

**Figure 5 diagnostics-11-01605-f005:**
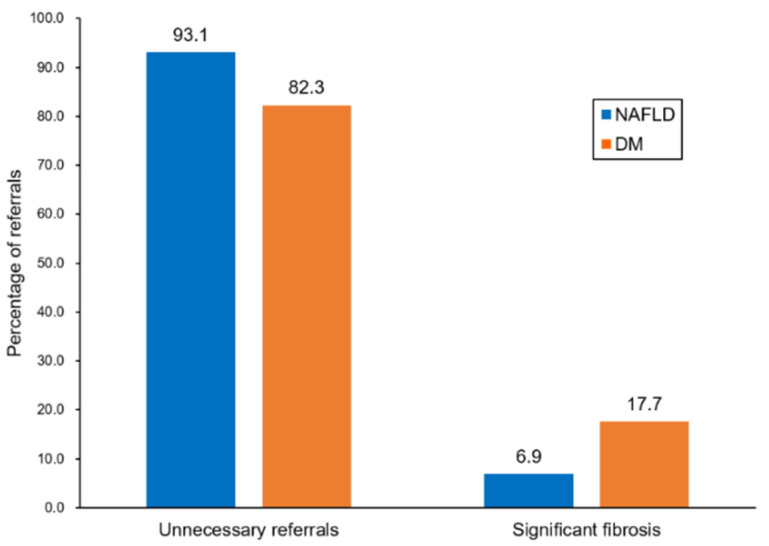
Evaluation of patients referred to secondary care.

**Table 1 diagnostics-11-01605-t001:** Baseline characteristics.

Characteristics	Male	Female	Total	*p*-Value
	*n* = 995	*n* = 293	*n* = 1288	
Age (years)	50.35 ± 8.33	51.28 ± 8.18	50.56 ± 8.30	0.094
Hypertension	349 (35.08)	69 (23.55)	418 (32.45)	<0.001
Diabetes	143 (14.37)	26 (8.87)	169 (13.12)	0.014
Impaired fasting glucose	382 (38.39)	73 (25.06)	457 (35.48)	<0.001
Metabolic syndrome	247 (24.82)	70 (23.89)	317 (24.61)	0.744
BMI (kg/m^2^)	24.88 ± 2.86	23.57 ± 3.57	24.58 ± 3.08	<0.001
Waist circumference (cm)	86.61 ± 8.07	81.06 ± 9.92	85.35 ± 8.83	<0.001
ALT (IU/L)	31.32 ± 22.02	22.84 ± 15.94	29.39 ± 21.09	<0.001
AST (IU/L)	26.31 ± 15.06	23.50 ± 10.62	25.67 ± 14.21	<0.001
GGT (U/L)	50.20 ± 64.75	375.08 ± 584.23	124.1 ± 278.75	0.342
Albumin (g/L)	4.70 ± 0.24	4.62 ± 0.24	4.68 ± 0.24	<0.001
Fasting glucose (mg/dL)	102.81 ± 19.05	96.93 ± 13.48	101.17 ± 18.10	<0.001
Platelets (10^9^/L)	239.77 ± 51.02	251.40 ± 52.05	242.41 ± 51.47	<0.001
Triglycerides (mg/dL)	145.61 ± 119.39	106.56 ± 57.79	136.73 ± 109.71	<0.001
HDL-cholesterol (mg/dL)	54.13 ± 14.40	61.79 ± 17.25	55.87 ± 15.43	<0.001
LDL-cholesterol (mg/dL)	131.64 ± 34.65	129.15 ± 33.56	131.07 ± 34.41	0.277
HbA1c (mmol/L)	5.74 ± 0.66	5.66 ± 0.53	5.72 ± 0.63	0.037
MRE ≥ 3.0 kPa	44 (4.42)	6 (2.05)	50 (3.88)	0.064
MRE ≥ 3.6 kPa	5 (0.50)	1 (0.34)	6 (0.47)	0.722
NFS ≥ −3.16	815 (81.91)	222 (75.77)	1037 (80.51)	0.02
FIB-4 index ≥ 0.92	578 (58.09)	170 (13.20)	748 (58.07)	0.969

Data are mean ± standard deviation or *n* (%) or med (IQR). BMI, body mass index; ALT, alanine transaminase; AST, aspartate transaminase; GGT, γ-glutamyltransferase; HDL, high-density lipoprotein; LDL, low-density lipoprotein; HbA1c, glycated hemoglobin; MRE, magnetic resonance elastography; NFS, NAFLD fibrosis score; FIB-4, Fibrosis-4.

**Table 2 diagnostics-11-01605-t002:** Performance of FIB-4 index and NFS based on various conditions.

Target Population	FIB-4 Index-Based Algorithm to Detect Significant Fibrosis (≥F2)				
	AUROC	Sensitivity (%)	Specificity (%)	PPV (%)	NPV (%)
USG-diagnosed NAFLD	0.570	42.0	77.2	6.9	97.1
Elevated liver enzyme	0.552	18.0	89.0	6.2	96.4
Metabolic syndrome	0.594	30.0	86.9	8.5	96.9
Impaired fasting glucose	0.594	24.0	78.3	4.3	96.2
Diabetes	0.688	40.0	92.5	17.7	97.5
**Target Population**	**NFS-Based Algorithm to Detect Significant Fibrosis (≥F2)**				
	**AUROC**	**Sensitivity (%)**	**Specificity (%)**	**PPV (%)**	**NPV (%)**
USG-diagnosed NAFLD	0.610	56.0	66.0	6.2	97.4
Elevated liver enzyme	0.499	26.0	84.8	6.5	96.6
Metabolic syndrome	0.521	46.0	78.8	8.0	97.3
Impaired fasting glucose	0.649	32.0	66.0	3.7	96.0
Diabetes	0.582	46.0	89.3	14.7	97.6

FIB-4, Fibrosis-4; NFS, NAFLD fibrosis score; AUROC, area under the curve of the receiver operating characteristic; PPV, positive predictive value; NPV, negative predictive value.

## Data Availability

The data presented in this study are available on request from the corresponding author. The data are not publicly available due to privacy.

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
