# Peer review of "Determining the Target Population That Would Most Benefit from Screening for Hepatic Fibrosis in a Primary Care Setting"

_diagnostics, 2021, doi:10.3390/diagnostics11091605_

Round 1

Reviewer 1 Report

This study reviewed a large amount of data from 1288 patients who underwent MRE in a health checkup, which is considered to be valuable clinical data. However, I have several questions about this study:

1) whether FIB4 and NFS, a scoring system originally proposed to pick up patients with advanced fibrosis, should be used for patients with significant fibrosis, as in this study;

2) Furthermore, in this study, the authors used cutoff points of 1.30 and -1.455 in FIB4 and NFS, respectively. However, these cutoff points were used for diagnosing advanced fibrosis, is the use of these cutoff points correct to diagnose the significant fibrosis as assessed by MRE ?

3) In Figure 1, it appears that 50 patients were selected in each group, but the rationale for this is not clear.

Author Response

This study reviewed a large amount of data from 1288 patients who underwent MRE in a health checkup, which is considered to be valuable clinical data. However, I have several questions about this study:

1) whether FIB4 and NFS, a scoring system originally proposed to pick up patients with advanced fibrosis, should be used for patients with significant fibrosis, as in this study;

2) Furthermore, in this study, the authors used cutoff points of 1.30 and -1.455 in FIB4 and NFS, respectively. However, these cutoff points were used for diagnosing advanced fibrosis, is the use of these cutoff points correct to diagnose the significant fibrosis as assessed by MRE ?

Author responses:

1,2) Thank you for your comment and we totally agree with you. As reviewer pointed out, FIB-4 and NFS were originally proposed to exclude advanced fibrosis using the cut-off values of 1.3 and -1.455, respectively. Thresholds should be different, as suggested by reviewer, to evaluate significant fibrosis among low-risk individuals. Therefore, we re-evaluated diagnostic performance and decision curve analysis using different cut-off values of 0.92 and -3.16 for FIB-4 and NFS, respectively (Ref. Obes Surg 2020, 30, 2538-2546, United European gastroenterology journal 2019, 7, 1124-1134). In the diabetes group, true positive rate of FIB-4 index was decreased at the chosen thresholds from 28.9% to 17.7%, whereas in USG-diagnosed NAFLD group, FIB-5 index showed similar result (7% to 6.9%). However, the overall flow of content has not changed much. The true positive rate and AUROC in diabetic groups was still higher than in the other groups including USG-diagnostic NAFLD group. We have changed tables and figures. These modified thresholds were not fully validated yet. Therefore, we have additionally described about this as a limitation of our study (Page9 , Line No.293-298).

3) In Figure 1, it appears that 50 patients were selected in each group, but the rationale for this is not clear.

Author responses:

3) Thank you for your comment. The number of patients with significant fibrosis with 3.0 kPa or higher based on MRE criteria was 50.

Reviewer 2 Report

The authors concluded that patients with diabetes, even in the absence of fatty liver, would be preferable for hepatic fibrosis screening with FIB-4 in a primary care setting. It is important to identify the appropriate population for hepatic fibrosis screening, thus the investigation seems to include meaningful contents. However, there are some critical issues to be addressed.

Major comments:

  1. Already in EASL clinical practice guidelines on non-invasive tests for evaluation of liver disease severity and prognosis – 2021 update, it is recommended that Non-invasive fibrosis tests should be preferentially used in patients at risk of advanced liver fibrosis (such as patients with metabolic risk factors and/or harmful use of alcohol). Of course, the patients included those with type 2 diabetes regardless of fatty liver. The authors should clearly describe the novelty of this study.
  2. It is agreeable that the cases with obvious high-risk for hepatic fibrosis, such as hepatitis virus positive or significant alcohol intake, were excluded. But USG is not always performed to all patients in the community setting. For the evaluation of the screening methods to rule-out patients with a low probability of having advanced fibrosis and prompt further testing for patients with indeterminate and positive results in the primary care clinic, patients with suspicious chronic liver disease on USG should be included in the study.
  3. Age is included in the formula of FIB-4. And the age of this study subjects seems to be relatively young compared with that of the patients in the primary care clinic. It needs to verify whether the results of this study apply to the elder population. The authors should describe it as a limitation of this study.

Author Response

The authors concluded that patients with diabetes, even in the absence of fatty liver, would be preferable for hepatic fibrosis screening with FIB-4 in a primary care setting. It is important to identify the appropriate population for hepatic fibrosis screening, thus the investigation seems to include meaningful contents. However, there are some critical issues to be addressed.

Major comments:

1. Already in EASL clinical practice guidelines on non-invasive tests for evaluation of liver disease severity and prognosis – 2021 update, it is recommended that Non-invasive fibrosis tests should be preferentially used in patients at risk of advanced liver fibrosis (such as patients with metabolic risk factors and/or harmful use of alcohol). Of course, the patients included those with type 2 diabetes regardless of fatty liver. The authors should clearly describe the novelty of this study.

Author responses:

1) Thank you for your comment. As the reviewers pointed out, it is very well known that the risk of hepatic fibrosis increased, when metabolic syndrome, including diabetes, combined. In addition, it is already known that screening for hepatic fibrosis is necessary in diabetes or NAFLD cohort. In this study, we tried to compare the social benefits of hepatic fibrosis screening among known various high risk groups (ex. USG-diagnosed NAFLD, liver enzyme abnormalities, metabolic syndrome, diabetes, etc.) in community cohort. Through benefit analysis from community base cohort, we tried to compare the size of social benefits among each high-risk group under head-to-head comparison. So far, a few social economic analysis was performed in selected high risk population with conflicting results, but the comprehensive net benefit analysis according to various high risk groups in primary care setting has not been reported. We believe that the results of this study can be helpful in understanding the target population and priority of screening algorithms in various high risk groups in primary care center. So, we added above comment as a discussion of our study (Page9 , Line No.251-259).

2. It is agreeable that the cases with obvious high-risk for hepatic fibrosis, such as hepatitis virus positive or significant alcohol intake, were excluded. But USG is not always performed to all patients in the community setting. For the evaluation of the screening methods to rule-out patients with a low probability of having advanced fibrosis and prompt further testing for patients with indeterminate and positive results in the primary care clinic, patients with suspicious chronic liver disease on USG should be included in the study.

Author responses:

2) We sincerely apologize for confusing you. All 27 patients with chronic liver disease on ultrasonography were all clinically definite cirrhosis patients. As you pointed out, low probability of having advanced fibrosis or patients with indeterminate group should be included. We excluded participants with pre-existing positive viral hepatitis or significant alcohol intake to evaluate the diagnostic performance of non-invasive tests in the "average risk" population. For the same reason, cases in which cirrhosis was clearly confirmed by ultrasound were excluded. Cases with low probability of chronic liver disease on ultrasonography were not excluded. In the flow chart of Figure 1 and methods section, chronic liver disease was corrected to liver cirrhosis (Page2, Line No. 84).

3. Age is included in the formula of FIB-4. And the age of this study subjects seems to be relatively young compared with that of the patients in the primary care clinic. It needs to verify whether the results of this study apply to the elder population. The authors should describe it as a limitation of this study.

Author responses:

3) Thank you for your comment and we agree with you. We have described about this as a limitation of our study (Page9, Line No.301-303).

Round 2

Reviewer 1 Report

The authors revised the paper well.  I think this paper is acceptable for "Diagnostics".

Reviewer 2 Report

The authors well responded point by point to the comments made by the reviewer.